# When Corticospinal Inhibition Favors an Efficient Motor Response

**DOI:** 10.3390/biology12020332

**Published:** 2023-02-20

**Authors:** Sonia Betti, Giovanni Zani, Silvia Guerra, Umberto Granziol, Umberto Castiello, Chiara Begliomini, Luisa Sartori

**Affiliations:** 1Department of Psychology, Centre for Studies and Research in Cognitive Neuroscience, University of Bologna, Viale Rasi e Spinelli 176, 47521 Cesena, Italy; 2Department of General Psychology, University of Padova, Via Venezia 8, 35131 Padova, Italy; 3School of Psychology, Victoria University of Wellington, Kelburn Parade 20, Wellington 6012, New Zealand; 4Padua Center for Network Medicine, University of Padova, Via Francesco Marzolo 8, 35131 Padova, Italy; 5Padova Neuroscience Center, University of Padova, Via Giuseppe Orus 2, 35131 Padova, Italy

**Keywords:** social interactions, transcranial magnetic stimulation, kinematics, motor-evoked potential, electromyography

## Abstract

**Simple Summary:**

Scientific evidence has demonstrated that in social contexts the motor system integrates predictions about the actions of others and plans the most appropriate responses. However, it is not yet known how these processes are reflected in the modulation of excitatory and inhibitory corticospinal mechanisms. Our goal was to reveal the behavioral and neurophysiological signatures characterizing different stages of these motor responses. Greater corticospinal inhibition was found when participants prepared their motor response after observing an interactive request compared with a non-interactive gesture, and this in turn favored a faster and more efficient execution of the action. With this neural finding, we shed light on the critical intermediate phase of motor processing between the observation of the action and the action integration phase aimed at achieving a joint goal. Taken together, these results will contribute to the understanding of motor processes that occur in social situations such as those we routinely encounter in our daily lives.

**Abstract:**

Many daily activities involve responding to the actions of other people. However, the functional relationship between the motor preparation and execution phases still needs to be clarified. With the combination of different and complementary experimental techniques (i.e., motor excitability measures, reaction times, electromyography, and dyadic 3-D kinematics), we investigated the behavioral and neurophysiological signatures characterizing different stages of a motor response in contexts calling for an interactive action. Participants were requested to perform an action (i.e., stirring coffee or lifting a coffee cup) following a co-experimenter’s request gesture. Another condition, in which a non-interactive gesture was used, was also included. Greater corticospinal inhibition was found when participants prepared their motor response after observing an interactive request, compared to a non-interactive gesture. This, in turn, was associated with faster and more efficient action execution in kinematic terms (i.e., a social motor priming effect). Our results provide new insights on the inhibitory and facilitatory drives guiding social motor response generation. Altogether, the integration of behavioral and neurophysiological indexes allowed us to demonstrate that a more efficient action execution followed a greater corticospinal inhibition. These indexes provide a full picture of motor activity at both planning and execution stages.

## 1. Introduction

In social contexts, the motor system integrates predictions about the actions of others and plans the most appropriate responses, with a very high level of sophistication [1,2,3,4,5]. To achieve an efficient level of coordination, observers must simulate the unfolding of an observed request, compare the upcoming sensory input, predict the partner’s intention from early movement cues [6,7,8,9,10] and finally activate the corresponding motor representation, well before the interactive request is fully expressed ([11,12] for a review). However, how such delicate integration is reflected in modulation of corticospinal excitatory and inhibitory mechanisms is still poorly studied. Transcranial magnetic stimulation (TMS) applied to the primary motor cortex and the resulting index, i.e., corticospinal excitability (CSE) is the technique of choice for studying motor system responses to observation of others’ actions (see [13] for a review). Although CSE modulations during passive observation of video clips have been extensively established, the investigation of the planning phase of motor interactions in realistic social situations is still very rare (see [14,15] for the investigation of inhibitory mechanisms during concurrent action execution and observation).

Moreover, the previous research has not fully disambiguated the relative contribution of two different mechanisms: action prediction (a priori expectations) and action integration (a posteriori processing of both partners’ actions). A recent study by Pesquita and colleagues [11] was able to isolate action prediction from confounding factors related to action integration. Specifically, participants watched animated sequences with two characters (e.g., A and B involved in a handshake sequence). Interestingly, judgments were facilitated when B’s reaction was temporally aligned with expectations about its performance (e.g., in the case of the handshake, when the arm was extended correctly and at the right time). This seems to suggest that participants were actively predicting rather than simply performing an a posteriori action integration. However, to advance the understanding of how humans perceive dynamic social interactions, the interdependencies between action prediction and action integration mechanisms still need to be clarified.

In the present study, we aimed to carefully decouple action prediction from action integration mechanisms (i.e., we avoided the possible effects of motor interference due to simultaneous observation and execution of an ongoing gesture) and extended the previous evidence based on the observation of video stimuli by adopting a real-time naturalistic setting in which participants were seated at the same table with a co-experimenter while performing meaningful actions. In particular, participants approached a coffee cup close to them and performed a precision grip (PG) on a spoon to stir the coffee or a whole-hand grasp (WHG) on the coffee cup to lift it after observing the co-experimenter trying to pour some sugar with a spoon in their coffee cups. These sequences of actions were selected on the basis of previous research showing that they were capable of eliciting in the observer the preparation of a response movement, even though in those studies no response was required as observers were simply asked to passively watch the actions [8,16,17,18,19]. In this respect, social signals can act as instrumental gestures, whose function is to alter a recipient’s behavior by triggering a range of opportunities for action [20,21]. Crucially, the end of the co-experimenter’s action served as a go signal, after which the participants had to release the start button and execute a predefined action (i.e., PG or WHG). This way, the two movements were decoupled: there was no temporal overlap between action observation and action execution, and no confound related to integration mechanisms was therefore present (i.e., no physical interaction ever occurred).

We recorded both response preparation and execution through a multimodal approach [22]. Action planning was measured by probing CSE and reaction times (RTs). Action execution was then calculated by means of electromyographic activity (EMG) and kinematics. Finally, participants filled out the Interpersonal Reactivity Index (IRI [23,24]) questionnaire to control for sample characteristics with respect to empathy-related dispositions. We capitalized on a previous robust paradigm (see [12] for a review) in which participants had to grasp the target object with a PG or a WHG. These actions are different in terms of muscular activations. Specifically, performing a precision grip entails the opposition of the thumb to the index finger, whereas a whole-hand grasp requires the opposition of all fingers to the palm, little finger included. Recording the CSE of first dorsal interosseous (FDI) and abductor digiti minimi (ADM) muscles allows then to investigate which motor representation is activated during the planning phase and to what degree.

Recently, Betti and colleagues [19] brought evidence of a CSE inhibition following observation of videos showing similar interactive requests directed to the participant. Notably, in that case participants had to passively watch the videos without responding. Two hypotheses were formulated to explain that inhibitory effect: (i) a selection mechanism because different motor representations were competing; or (ii) a muscular deactivation to prevent overt reactions (i.e., impulse control).

In this experiment, we adopted a block design to rule out the first hypothesis: participants were clearly instructed what action to perform and we adopted a delayed go signal to test the second hypothesis. We reasoned that if motor inhibition was still present, then it should be ascribed to an impulse control mechanism [25,26], a byproduct of increased preactivation due to social engagement. More specifically, if the observation of an interactive request strongly triggers the preparation of an appropriate response action (i.e., Social Motor Priming [27]), which, however, must be postponed, then more inhibition should occur for the interactive compared to the non-interactive condition. In this case, corticospinal inhibition acts as a dam to curb the impulse to respond, i.e., to perform the action automatically triggered by the social request. Accordingly, we hypothesized that CSE should diminish after observing the co-experimenter asking for an interactive response compared to observing a non-interactive gesture. In addition, we tested the action execution component by probing hand kinematics and muscular activity when participants actually performed the delayed response action. Altogether, the integration of CSE, RTs, EMG as well as hand kinematics allowed us to highlight that a more efficient action execution followed greater levels of CSE inhibition.

## 2. Materials and Methods

### 2.1. Participants

Twenty-eight volunteers (15 women and 13 men, aged between 19 and 30 years, mean age 23.89 ± 2.6 years) participated in the experiment. All participants were right-handed, as assessed with the Edinburgh Handedness Inventory [28], with normal or corrected-to-normal visual acuity. They were all screened for the TMS exclusion criteria and for neurological, psychiatric and medical problems [29,30]. The study was conducted in accordance with the Declaration of Helsinki, and approved by the Ethics Committee of the University of Padova (protocol n. 2371). All participants were naïve to the purposes of the experiment and gave their written informed consent for their participation.

### 2.2. Experimental Paradigm

The participants sat comfortably in front of a table (90 × 90 cm) on which was placed a cup (12 cm height, 9 cm diameter) full of coffee, with a teaspoon (20 cm long) inside it. The cup was placed in the middle sagittal plane of the participants, 30 cm from the edge of the table. On the other side of the table, a co-experimenter (female, 23 years) was seated in front of the participant, with a sugar spoon close to her right hand and a sugar bowl placed on her left. Both the participant and the co-experimenter placed their right hands on a starting button at the beginning of each trial (Figure 1A).

Participants were then tested under four experimental conditions (two interactive requests, two non-interactive actions) which were validated by a preliminary study designed to test for social appropriateness (see Appendix A):Interactive request, precision grip execution: the co-experimenter grasped the sugar spoon, took some sugar from a sugar bowl, and then stretched out her arm toward the participant’s cup, as if to pour the sugar in it. Participants had to reach and grasp their teaspoon with a PG, and to stir the coffee;Non-Interactive action, precision grip execution: the co-experimenter grasped the sugar spoon, took some sugar from the sugar bowl and then returned to the initial position. Participants had to reach and grasp their teaspoon with a PG, and to stir the coffee;Interactive request, whole-hand grasp execution: the co-experimenter grasped the sugar spoon, took some sugar from a sugar bowl, and then stretched out her arm toward the participant’s cup, as if to pour some sugar in it. Participants had to reach and grasp the cup with a WHG and to lift it up (i.e., bringing it closer to the teaspoon full of sugar);Non-Interactive action, whole-hand grasp execution: the co-experimenter grasped the sugar spoon, took some sugar from a sugar bowl and then returned to the initial position. Participants had to reach and grasp the cup with a WHG and to lift it up.

Given the important role played by the agent’s gaze, along with limb movements, in providing a source of information for social interaction [18,19,31], we carefully controlled for the co-experimenter’s gaze direction at different stages of the observed action and experimental conditions. Because previous research suggests that gaze toward the observer at the end of the action (i.e., direct gaze) increases perceived social engagement more than gaze toward the target object for the interaction [18], the co-experimenter was instructed to follow her hand movements in a natural way with gaze and, at the end of the movement, to look at the participant. In this way, the co-experimenter’s gaze was comparable in both the Interactive and Non-Interactive conditions.

### 2.3. Validation Study

Fifty naïve volunteers (36 women and 14 men, aged 18–74 years, mean age 28.41 ± 12.93 years) were recruited to validate the adopted action sequences in a preliminary study (see Appendix A). They watched four video clips showing a right-handed non-professional actress in a third-person perspective performing the above described gestures toward a partner presented in a first-person perspective. The videos were shot so that only the partner’s right hand was visible in the bottom right corner, as if the observer was actually participating in the scene (see Appendix A). After each video presentation, participants were requested to express their ratings on a five-point Likert scale (ranging from “not at all” to “very much”) for each item. Item Q1: “Is the girl’s action interactive?”; Item Q2: “Do I feel involved in the interaction?”; Item Q3: “Does the partner in the first person view correctly reply to the girl’s gesture?” (for a similar approach, see [18]). Interactive videos obtained statistically significant higher scores for all items (Q1, Q2, Q3) as compared to Non-interactive videos. Moreover, for the Interactive condition, 73% of participants considered the action of “lifting the cup” (WHG) to be more socially appropriate with respect to “stirring the coffee” (PG). We decided to keep actions with varying degrees of social appropriateness (i.e., how appropriate an action is in response to the observed action) to test whether or not corticospinal responses reflected this explicit evaluation [32]. For a detailed report of the methods and obtained results see the Appendix A.

### 2.4. Procedure

Participants were individually tested in a single experimental session lasting approximately two hours. They were seated in a comfortable chair with their right elbow positioned on an adjustable armrest and the ulnar styloid process of their right hand on the starting platform for the recording of reaction times. The resting hand was kept relaxed and in a pinch position with two pairs of surface EMG electrodes and two infrared markers for kinematic analyses. The participant’s head was supported by a fixed headrest and a TMS coil was placed on their head (Figure 1). Participants observed the co-experimenter’s Interactive or Non-Interactive request action and at its conclusion they received a TMS pulse for MEP recordings, which also served as a go signal. The time of administration of the TMS was determined on the basis of the duration of the co-experimenter’s gesture (i.e., 5 s) so that the two events (end of gesture and onset of the pulse) coincided in all the conditions. Afterwards, the reaction times, hand kinematics and EMG activity of the participants were simultaneously recorded while they were performing either a precision grip on the spoon or a whole-hand grasp on the coffee cup.

At the end of each trial, participants were requested to move the object back to its original position and then return to their start button. They performed four blocks of 16 trials. During each block they had to perform the same action (i.e., PG or WHG) that was inverted in the subsequent block. The type of action to perform was verbally indicated at the beginning of each block, and the order was counterbalanced between participants. Between each trial, they were instructed to remain as still and relaxed as possible. In addition, sixteen baseline trials were acquired both at the beginning and at the end of the experimental session: participants performed the same type of action (PG/WHG) for eight consecutive trials (i.e., Baseline PG, Baseline WHG), while the co-experimenter sat still at the table, to control for possible audience effects due to the presence of another individual [33]. The participants conducted a total of 96 trials: 32 baseline and 64 task trials; overall, they performed 16 repetitions for each type of action and condition. After taking part in the experiment, participants completed the Italian version of the IRI [23,24].

### 2.5. Kinematic Recording

Movements were recorded using a 3-D optoelectronic SMART-D system (Bioengineering Technology and Systems, B|T|S|) equipped with six infrared cameras (sampling rate 60 Hz), placed in a semicircle at a distance of 1 to 1.2 m from the table (Figure 1A). Two semi-spherical reflecting markers (~0.25 mm diameter) were attached to the participant’s right hand, namely, on the radial side of the index nail and on the ulnar side of the thumb nail (Figure 1E). These markers served to measure the manipulation component [34,35]. Static and dynamic calibrations were performed for 3-D space reconstruction as described in [22]. The standard deviation of the reconstruction error was below 0.3 mm for all axes (x, y, z).

To check for possible effects of the co-experimenter’s action execution on the participant’s response (e.g., [36]), a semi-spherical reflective marker was also attached to the radial styloid process of her wrist to measure its velocity profile across different experimental conditions. Because we wanted to test the possible influence of interactive requests on the participant, we considered the speed of the wrist in the concluding phase of the sequence (i.e., from when it rises on the sugar bowl and in one case heads toward the participant’s cup, whereas in the other case it moves back to the starting position). We did not consider the first phase of reaching for the teaspoon nor the object manipulation component because they were exactly the same across conditions (i.e., the teaspoon was always placed in the same spot and the thumb and index finger were always closed on the object).

### 2.6. Electromyography

Surface EMG activity was recorded simultaneously from the first dorsal interosseous (FDI) and abductor digiti minimi (ADM) muscles of the participant’s right hand through two pairs of Ag/AgCl electrodes (1 cm diameter) placed in a belly-tendon montage (Figure 1B). After the skin was cleaned, electrodes with a small amount of water-soluble EEG conductive paste were placed and fixed on the target positions. The active electrode was placed over the belly of the muscle, determined by palpation during maximum voluntary contraction, and the reference electrode was placed over the proximal interphalangeal juncture. The ground electrode was placed on the participant’s right wrist. The electrodes and wires were secured and placed so as not to restrict the participant’s movements. The skin impedance, evaluated at rest prior to beginning the experimental session, was considered of good quality when below the threshold level (5 kOhm). The electrodes were connected to an isolable portable ExG input box linked to the main EMG amplifier for signal transmission via a twin fiber optic cable (Professional BrainAmp ExG MR, Munich, Germany). A high-pass filter of 30 Hz and a low-pass filter of 1000 Hz were applied to the raw myographic signal, which was amplified prior to being digitalized (5 kHz sampling rate), and stored on a computer for off-line analysis. Due to technical problems during data acquisition, for one participant, the EMG data file acquired during action execution in the task phase was corrupted and no further analysis was performed. EMG signals were recorded with Brain Vision Recorder software (Brain Products GmbH, Munich, Germany).

### 2.7. Transcranial Magnetic Stimulation

Single-pulse TMS was administered using a 70 mm figure-of-eight coil connected to a Magstim BiStim^2^ stimulator (Magstim Co., Whitland, UK). Pulses were delivered to the left primary motor cortex (M1) of the participant, in correspondence with the representation of the right hand (Figure 1C). The coil was placed on the head at a 45-degree angle relative to the interhemispheric fissure, with the handle pointing laterally and caudally [37,38]. The optimal scalp position (OSP), which is defined as the best position for the coil on the scalp at which the lower intensity of stimulation elicits the largest MEP (Figure 1D) of both ADM and FDI muscles, was determined by moving the coil in approximately 0.5 cm steps around the presumed hand motor area. The OSP was then marked on a tight-fitting cap worn by the participants, ensuring a correct coil placement throughout the experiment. During the experiment, the coil was held on a tripod and the experimenter continuously checked the position to maintain a constant positioning with respect to the marked OSP. For each participant, the resting motor threshold (rMT), that is, the lowest stimulation intensity inducing MEPs with at least ≥50 µV peak-to-peak amplitude in a relaxed muscle was found in 50% of 10 trials [39] in the muscle of higher-threshold. rMT ranged from 28 to 52% (mean = 40.6%, SD = 5.68) of the maximum stimulator output. The stimulation intensity was then set at 120% of the rMT. TMS stimulation and EMG recording were managed by E-Prime V2.0 software (Psychology Software Tools Inc., Pittsburgh, PA, USA).

### 2.8. Interpersonal Reactivity Index Questionnaire

The Interpersonal Reactivity Index (IRI [23,24]) assesses empathy-related dispositions, and is made by 28 items divided in four subscales evaluating different aspects of empathy: Perspective Taking (PT, the dispositional tendency to adopt the perspective of another individual); Empathetic Concern (EC, the tendency to experience feelings of sympathy and compassion for others in need); Personal Distress (PD, the tendency to feel distress and anxiety as a result of witnessing another’s emotional distress); and Fantasy Scale (FS, the tendency to transpose themselves imaginatively into the feelings and actions of fictional characters and situations). Five participants did not return their responses to the questionnaire and were therefore not included in the analysis.

### 2.9. Data Preparation

#### 2.9.1. MEP Data

Individual peak-to-peak MEP amplitudes (mV) were analyzed offline using Brain Vision Analyzer (Brain Products, BmbH, Munich, Germany). The peak-to-peak MEP amplitude for the FDI and ADM muscles was determined as a measure of the corticospinal excitability of the participants. Trials in which EMG activity greater than 100 µV was present in the 100 ms window preceding the TMS pulse were discarded to prevent contamination of the MEP measurements by background EMG activity (<1%); in addition, values greater than 3 SD of the mean were excluded as outliers (<5%). To control for inter-individual variability in MEP amplitudes, separately for each participant and each muscle, the raw MEP amplitudes (including baselines) were z-transformed.

#### 2.9.2. EMG Data

The EMG activity was analyzed offline using Brain Vision Analyzer (Brain Products BmbH, Munich, Germany). The EMG signal from the FDI and ADM muscles during action execution was rectified and the area under the curve of the rectified EMG track (mV*s) was calculated for each trial and for each muscle to quantify muscle activity when executing grasping actions (i.e., PG, WHG). As for the MEP, separately for each participant and each muscle, the raw EMG data were z-transformed. EMG activity was measured within a time window starting 500 ms after the TMS-go signal pulse up to 4000 ms.

#### 2.9.3. Reaction Times

RTs were collected as the time delay between the go signal and the time at which participants released the start button.

#### 2.9.4. Kinematic Data

Following infrared marker data collection, their 3-D positions were reconstructed as a function of time, filtered (Butterworth filter with a 6 Hz cut-off) and analyzed using the SMART-D Tracker and SMART-D Analyzer software packages (B|T|S). Given that we aimed to investigate the crosstalk between finger muscles’ activity and their kinematics during the prehension of different objects (see [22] for a similar approach), analyses were focused on the grasp component (that is, finger preshaping when approaching the object and finger closing around it [34,35]). The following kinematic parameters were then extracted for each individual movement to measure the manipulation component (e.g., [40]).

Grasping Time (GT), the time interval between grasping onset (i.e., the time at which the hand contacted the object, quantified as the time at which the hand opening velocity crossed a threshold of 5 mm/s after reaching its minimum value and remained above it for longer than 500 ms) and grasping offset.

Time to Maximum Grip Aperture (TMGA), the time spent to reach peak grip aperture, that is, the maximum distance by the 3-D coordinates of the thumb and index finger (ms).

As concerns the co-experimenter, we extracted the peak wrist velocity (PWV, i.e., the maximum velocity reached by the wrist), while she was stretching out her arm toward the out-of-reach cup (Interactive condition), or when returning the spoon to the initial position (Non-Interactive condition).

#### 2.9.5. Interpersonal Reactivity Index Questionnaire

The scores for the four subscales of the IRI questionnaire were in line with those reported by [41], with a mean (±SD) score of 17.52 (±2.68) for the EC subscale, of 17.88 (±4.33) for the FS, 11.8 (±4.8) for the PD and 19 (±3.81) for the PT subscale.

### 2.10. Statistical Analysis

The analyses were performed using the STATISTICA software (StatSoft Inc., version 8, Tulsa, OK, USA). For the MEP and EMG data, a repeated-measure ANOVA (rmANOVA) was performed with muscle (FDI, ADM), condition (Non-Interactive, Interactive) and type of grasp (PG, WHG) as within-subject factors. In addition, to test for any basal change in CSE during the experiment, the raw MEPs acquired during the initial and final baseline blocks during PG or WHG preparation were compared through paired-sample *t*-tests. For reaction times and kinematic data, separate rmANOVAs with condition (Non-Interactive, Interactive) and type of grasp (PG, WHG) were performed for each parameter. The partial eta square (η^2^_p_) value was used as an estimate of effect size. A significance threshold of 0.05 was set for all statistical analyses. In the presence of significant interactions, post-hoc comparisons were performed using the Bonferroni correction. For each variable, planned orthogonal contrasts were analyzed to test the comparison between Interactive and Non-Interactive actions for each type of grasp. Finally, Pearson’s correlations were computed to explore the possible relationship between action preparation and action execution measures based on the interactive context. For this purpose, we calculated a differential index obtained by subtracting the Interactive to the Non-Interactive condition for each variable (i.e., raw MEP, RTs, GT and TMGA delta values). According to our hypothesis, if Interactive action is characterized by a stronger CSE inhibition and more efficient action execution, we should find a positive relationship between our delta indexes.

## 3. Results

### 3.1. Motor Preparation

#### 3.1.1. Motor-Evoked Potentials

No significant differences emerged when comparing raw MEPs recorded during the initial and final baseline blocks, either for FDI and ADM during PG (t_28_ = 1.49, *p* = 0.15 and t_28_ = 1.59, *p* = 0.12, respectively) and WHG preparation (t_28_ = −0.005, *p* = 0.99 and t_28_ = 1.49, *p* = 0.15, respectively). This indicates that TMS per se had not induced general changes in motor excitability during the experiment.

The rmANOVA performed on the normalized MEPs (z-scores; see Figure 2) revealed a main effect of condition (F_1,28_ = 15.32, *p* < 0.001, η^2^_p_ = 0.35), with lower MEP amplitudes in the Interactive than the Non-interactive condition. A significant interaction muscle x grasp (F_1,28_ = 16.46, *p* < 0.001, η^2^_p_ = 0.37) emerged, having increased MEP amplitude for the FDI muscle in PG than in WHG preparation (*p* < 0.001) and increased MEP amplitude for the ADM muscle in WHG preparation than the FDI muscle (*p* = 0.014). Interestingly, also a significant interaction muscle x condition (F_1,28_ = 9.07, *p* = 0.006, η^2^_p_ = 0.25) emerged, with post-hoc contrasts showing a greater MEP inhibition for the Interactive than the Non-interactive condition during motor preparation in both the FDI (*p* < 0.001) and the ADM (*p* = 0.004) muscles. In addition, the MEP amplitude for the FDI muscle in the Interactive condition was lower than the ADM muscle both in the Non-interactive (*p* < 0.001) and the Interactive (*p* = 0.03) conditions, with the latter being significantly reduced with respect to the FDI muscle in the Non-Interactive condition (*p* < 0.001). No other main effects nor interactions were significant (all F < 3.23, *p* > 0.09, η^2^_p_ < 0.12). The planned contrasts showed that the differences between Interactive and Non-Interactive conditions for the FDI muscle emerged for both PG and WHG preparation (t_28_ = 3.41, *p* = 0.002 and t_28_ = 2.76, *p* = 0.01, respectively), whereas the reduction in CSE in the Interactive condition was specific for the WHG action preparation (t_28_ = 2.63, *p* = 0.013), but not the PG (t_28_ = 1.25, *p* = 0.22), when considering the ADM muscle.

#### 3.1.2. Reaction Times

Regarding the reactivity to the observed actions, the rmANOVA on RTs showed a main effect of condition (F_2,54_ = 14.47, *p* < 0.001, η^2^_p_ = 0.35; Figure 3). The longest RTs were measured for the Baseline condition (724.14 ± 161.79 ms) compared to both the Non-Interactive (653.40 ± 147.14 ms) and Interactive (627.50 ± 137.19 ms) conditions (*p* = 0.001 and *p* < 0.001, respectively). No other main effects nor interactions were significant (all F < 0.65, *p* > 0.43, η^2^_p_ < 0.02). The planned contrasts showed that the Interactive and the Non-Interactive conditions differ significantly, with lower RTs for the Interactive condition both when PG (t_27_ = 2.65, *p* = 0.013) and WHG (t_27_ = 2.12, *p* = 0.044) actions are performed.

### 3.2. Motor Execution

#### 3.2.1. Electromyography

As for motor execution, the rmANOVA showed a statistically significant interaction between muscle and grasp (F_1,27_ = 166.41, *p* < 0.001, η^2^_p_ = 0.86; Figure 4). Classical differences in muscular activity characterizing the two types of grasp emerged, in particular, ADM was more activated for WHGs than PGs (*p* < 0.001), and FDI was more activated for PGs than WHGs (*p* < 0.001). Furthermore, PGs were characterized by greater participation of FDI than ADM (*p* < 0.001), and WHGs by a greater participation of ADM than FDI (*p* < 0.001). The planned contrasts for each muscle showed no differences between Interactive and Non-Interactive conditions for both the PG (FDI: t_27_ = 0.31, *p* = 0.76; ADM: t_27_ = −0.10, *p* = 0.92) and WHG actions (FDI: t_27_ = 1.88, *p* = 0.07; ADM: t_27_ = −0.74, *p* = 0.47).

#### 3.2.2. Kinematics

Grasping time (GT). The rmANOVA on GT data (see Figure 5A) showed a main effect of grasp (F_1,27_ = 145.18, *p* < 0.001, η^2^_p_ = 0.84) as well as of condition (F_1,27_ = 7.46, *p* = 0.001, η^2^_p_ = 0.22). In particular, a longer GT characterized the coffee mug-lifting execution (WHG) compared to stirring action (PG). More interestingly, the longest GT was measured for the Baseline condition (1022.67 ± 391.28 ms) compared to both the Non-Interactive (987.98 ± 366.63 ms) and the Interactive (966.39 ± 362.75 ms) conditions (*p* = 0.065 and *p* = 0.001, respectively), but the GT reduction was significant only for the Interactive condition. However, the interaction between grasp and condition was not significant (F_2,54_ = 0.84, *p* = 0.44, η^2^_p_ = 0.03). The planned contrasts showed that the Interactive and the Non-Interactive conditions differ significantly, with a reduced GT for the Interactive action when executing a PG (t_27_ = 2.45, *p* = 0.02) but not a WHG (t_27_ = 1.11, *p* = 0.28) action.

Time to Maximum Grip Aperture (TMGA). The rmANOVA on TMGA (see Figure 5B) showed a main effect of grasp (F_1,27_ = 195.27, *p* < 0.001, η^2^_p_ = 0.88) and condition (F_2,54_ = 11.27, *p* < 0.001, η^2^_p_ = 0.29). The peak of maximum grip aperture occurred earlier when participants grasped the sugar spoon (PG) as compared to the coffee mug (WHG). As regards the main effect of condition, the TMGA occurs later in the Baseline (652.52 ± 353.97 ms) than the Non-Interactive (617.98 ± 322.97 ms) and Interactive (610.57 ± 329.88 ms) conditions (*p* = 0.002 and *p* < 0.001, respectively). The interaction between grasp and condition was not significant (F_2,54_ = 2.98, *p* = 0.06, η^2^_p_ = 0.1). The planned contrasts showed that the Interactive and the Non-Interactive conditions differ significantly, with an anticipated peak of MGA for the Interactive condition for PG (t_27_ = 2.39, *p* = 0.02) but not WHG (t_27_ = −0.12, *p* = 0.91) actions.

Co-Experimenter’s peak wrist velocity (PWV). From the rmANOVA, when considering the PWV for the actions performed by the co-experimenter, only a main effect of the condition emerged (F_1,27_ = 69.31, *p* < 0.001, η^2^_p_ = 0.72). A higher peak of wrist velocity emerged for the Non-Interactive than the Interactive actions, characterizing the two different movements the actress performed (i.e., approaching the out-of-reach object vs. returning to the initial position). However, no main effect of grasp (F_1,27_ = 2.43, *p* = 0.13, η^2^_p_ = 0.08) nor a grasp x condition interaction (F_1,27_ = 2.35, *p* = 0.14, η^2^_p_ = 0.08) emerged. This suggests that the maximum velocity reached by the actress’ wrist for Interactive and Non-Interactive conditions showed a comparable peak velocity for both the PG and WHG trials. It is then possible to rule out that differences in the co-experimenter’s actions between conditions that could have influenced the subsequent participant’s action.

### 3.3. Correlations

To test the relationship between action preparation and action execution phases based on the social context, Pearson’s correlations were computed on differential indexes obtained by subtracting the Interactive to the Non-Interactive condition. The results show a positive correlation between MEP and RT data for both the FDI and ADM muscle for WHGs (r = 0.51, *p* = 0.005 and r = 0.41, *p* = 0.029, respectively; Figure 6A,B), namely, stronger inhibition for Interactive than Non-Interactive conditions is positively associated with lower RT for Interactive than Non-Interactive conditions. No significant correlations with MEP emerged for PG actions (FDI: r = 0.25, *p* = 0.19; ADM: r = 0.28, *p* = 0.15), nor for GT and TMGA indexes of action execution (all *p* > 0.05).

## 4. Discussion

In the present study, we evaluated CSE, EMG, RTs and dyadic 3-D kinematics during the preparation and execution of hand movements after observing a social request. This multimodal approach allowed us to investigate the crosstalk between CSE modulations and the subsequent motor execution, taking advantage of both the muscle specificity and good time resolution of MEP modulations, as well as of the fine-grained description of movement execution provided by the EMG and 3-D motion analysis. We aimed to reveal the behavioral and neurophysiological signatures characterizing different stages of a motor response in contexts calling (or not) for an interactive action. Greater motor inhibition was found when participants prepared their motor response after observing an interactive request compared to a non-interactive gesture. That is, even though participants were instructed to complete the same pre-determined action in each block, we still found differences based on whether participants had just observed an interactive or non-interactive action. This motor inhibition effect during the action preparation following observation of the interactive gesture confirmed the findings from a previous study based on passive observation of video stimuli [19] and identified its underlying mechanisms. Extending this line of research, here we also observed that in real-time scenarios, reduced MEPs during action preparation were followed by efficient performance during actions’ execution, in terms of faster and more efficient motor responses as shown by reaction times and movement kinematic.

### 4.1. Social Motor Inhibition

Previous investigations of the CSE index during passive observation of interactive requests have reported conflicting results showing both facilitatory [9,17,18,42] and inhibitory effects [19]. It is known that the willingness to engage in socially meaningful situations [43] increases the CSE index for the muscle involved in the appropriate motor response during an early time window (i.e., at the very beginning of an observed request gesture [18]). On the other hand, the more participants declare to be involved in a social interaction, the lower is their late-motor inhibition (i.e., at the end of a request gesture [19]). These findings are in fact consistent with the extensive literature on action anticipation, showing greater motor facilitation when observing the early stages of an action rather than its conclusion [44,45,46,47]. Naish et al. [13] developed a model to describe corticospinal excitability modulations in action observation studies. They suggest that, to prevent the overt imitation of the observed movements, inhibitory processes would follow an early increase in corticospinal excitability. The authors propose that these inhibitory mechanisms might occur either (or both) in parallel with excitatory processes, or might be triggered when the level of excitation reaches a certain threshold. During action observation tasks, muscular inhibition seems to be a necessary mechanism to prevent unwanted overt reactions. This might be particularly true for social requests, since they tend to activate prompt and uncontrollable responses, regardless of the given instructions [48]. The model from Naish and colleagues [13] could therefore be extended to situations in which the observed action not only involves motor resonance mechanisms, but also triggers in the observer the preparation of a response action. In particular, when the observation of an interactive request strongly triggers the preparation of an appropriate response action, an inhibition of corticospinal excitability is expected to prevent an overt behavior when a delayed response is required. Strong late inhibition can also be interpreted as a ‘rebound effect’: the more the motor system is activated, the more it should be subsequently inhibited [49]. Here, participants were requested to refrain from performing the response action until the go signal was presented. The results of the present study seem to confirm that the timing of TMS stimulation plays a key role, as CSE inhibition appears during a late time window both during the passive observation of interactive video clips [19] and in real social scenarios.

Corticospinal suppression during response preparation might reflect two independent inhibitory mechanisms. Since motor behavior results from constant competition between potential actions [50], multiple motor options are simultaneously activated until the activity associated with a particular action reaches a given threshold [51]. Response selection then entails inhibition of deselected muscles to avoid unwanted movements (i.e., inhibition for ‘deselection’ or ‘competition resolution’). Consistent with this view, a large number of studies have shown a decrease in CSE of non-selected hand muscles during movement preparation [52,53,54,55,56]. Interestingly, corticospinal suppression has also been shown to reflect another inhibitory mechanism [57,58], namely impulse control. ‘Impulse control’ inhibition is necessary to prevent selected actions from being performed prematurely (e.g., when the task requires withholding movement [59,60,61]). In general, the functional role of preparatory inhibition could be to reduce noise and enhance signal processing before response initiation [62]. In particular, previous studies suggest that decreases in CSE seem to go parallel with increases in M1 activity [63,64]. This further confirms the hypothesis of mechanisms that suppress corticospinal output at the spinal level by ensuring that movements are not prematurely initiated during strong cortical activation (i.e., a braking mechanism [57]).

The present study confirms and extends the previous literature on corticospinal inhibition that occurs when movement is prevented: after passive observation of motor actions [19] or during motor imagery [65]. Here, we adopted a real-time scenario and a customized integration of specific neurophysiological and behavioral indexes. In line with the ‘impulse control’ hypothesis, we demonstrated that a motor inhibition, as measured through MEP responses, occurs right before the go signal. This inhibition is specifically modulated by the presentation of the interactive request. Indeed, when participants were requested to grasp and lift the coffee cup following the co-experimenter’s request action, compared to the non-interactive action, a stronger CSE inhibition was associated with a quicker action initiation (i.e., shorter reaction times), as revealed by the correlation results. However, this effect emerged only for the action of lifting and not for the action of stirring, that is, for the action considered more appropriate in response to the interactive request gesture according to the validation study. Crucially, our behavioral data seem to entail that a ‘social motor inhibition’ is beneficial to the following performance of an appropriate gesture as emerged from reaction time and kinematic results. Recently, original neurophysiological findings during dyadic tasks showed that the delicate negotiation of interactive motor performance is best characterized by the fine-tuning of motor inhibition rather than excitation [14].

### 4.2. Social Motor Priming

Our reaction times and kinematics’ results confirm the facilitatory effect of acting after having observed a biological movement (i.e., motor priming [66,67]). In addition, our results confirm and extend the previous literature on social action observation [27,68] by showing that observing an interactive request in real time can facilitate the following execution of socially appropriate responses. In particular, reaction times—which reflect the preparation stage of the action—for the Interactive condition were shorter compared to both the Non-Interactive and Baseline conditions. Moreover, the grasping component of the Interactive precision grip movement was characterized by a longer closing of the hand while approaching the spoon, as indicated by the earlier time of maximum grip aperture in the Interactive compared to the Non-Interactive condition [48,69,70]. This indicates that participants chose an anticipation strategy and allocated more time to correctly calibrate fingertip placement on the object and to firmly finalize the grasping movement for the Interactive condition. Notably, this movement was also performed within a shorter time window of grasping time with respect to the Non-Interactive condition. This suggests that the movement was overall quicker and better calibrated, prompting the individual to respond efficiently to the interactive request.

Long-standing scientific evidence has shown that the motor response is facilitated if the action to be performed has been recently observed: this visuomotor priming effect is defined as the facilitation of performing an action that is congruent with the observed one [71]. Here, we found a consistent effect of visuomotor priming both for reaction times and kinematic variables (i.e., grasping time and time to maximum grip aperture) in the precision grip conditions after observing the co-experimenter grasping the sugar spoon (i.e., a PG) with respect to the baseline condition. This effect was enhanced when comparing the Interactive versus the Non-Interactive conditions, thus suggesting the presence of a Social Motor Priming [19]. Social contexts can indeed modulate motor performance [72,73], so that response execution can be speeded up compared to an imitative action [74,75].

No differences were found when performing a whole-hand grasp on the cup full of coffee in the Interactive and Non-Interactive conditions for any kinematic variable. This was likely due to the inherent difficulty of lifting a heavy cup without spilling the coffee. This ecological but challenging condition probably increased the accuracy level, bringing a general ceiling effect. This is confirmed by the temporal aspect of the kinematic variables: grasping time and time to maximum grip aperture were double when performing a WHG with respect to a PG (see Figure 5). This finding could be a useful warning for future ecological studies adopting real-life stimuli. In addition, muscle activations also do not appear to differentiate between the same action executed under Interactive and Non-Interactive conditions, suggesting that the EMG index may fail to reveal subtle changes in performance compared to the more detailed kinematic analysis of movement. This highlights the importance of acquiring different measures to investigate a phenomenon through different levels of sophistication.

In a previous study from our lab [76], we demonstrated that during whole-hand grasps the index finger tends to lift from the surface of the stimulus during demanding conditions. This strategy allows for greater control (i.e., a stabilizing mechanism) when stimulus dynamics become increasingly difficult (see also [77]). As recently confirmed by the authors of [22], the index finger can be regarded as a “navigator” during computation of a hand trajectory toward a target, even for whole-hand grasps. Careful placement of the digits driven by the index finger is considered a prerequisite for a stable grasp [78,79]. The interactive whole-hand grasp condition in our experiment, as also suggested by the results of the preliminary validation (see Appendix A), appeared to be effective in triggering an interactive response. However, the physical constraint provided by the liquid content of the object and its weight perhaps hindered this effect.

It is important to note that we excluded possible effects due to the variability of the co-experimenter’s movements on the subsequent participants’ response. Recent research has indeed shown that the way an action is performed (e.g., “vitality forms” [80]) might affect the subsequent partner’s response [36]. In the present study, the co-experimenter might have shown anticipatory mechanisms regarding the participant’s upcoming response (e.g., [2,81]). As demonstrated by the velocity results for the co-experimenter’s wrist, we ruled out this hypothesis by showing that she always performed the Interactive and Non-Interactive action sequences in the same way, regardless of the subsequent response action. Therefore, we can exclude that participants’ actions might have been affected by the co-experimenter’s movement variability.

### 4.3. The Encoding of Incomplete Joint Actions

The MEP results highlighted an inhibitory effect on motor preparation both when preparing a lifting and a stirring response following the interactive request. We hypothesize that participants implicitly filled the temporal gap between observed request and executed action, and mentally represented the joint goal of the action. During social interactions, the ability to read intentions from incomplete and initial movements reflects a predictive attitude of our motor system (‘predictive coding’ [31,82]), which is particularly relevant in ambiguous situations (e.g., [83]). In terms of CSE, the tuning to anticipatory simulation of observed actions appears, for example, when observers watch a hand approaching an object, with MEPs increasing before the hand contacts the object [45,46,47]. In more ecological and complex scenarios, predictive corticospinal activations emerged in social [8] and sporting contexts (e.g., [9,44]), where predicting and anticipating the effect of the observed action allowed for prompt and efficient reactions. In general, the social world is perceived as a continuum, although we only see the initial parts of the actions of others when we start to interact with them. Similar to the Gestalt principles in perception, in social contexts, humans might perceive other individuals in connection with objects and environments as part of a whole system of dynamic relationships. It is widely known that our visual system prioritizes the processing of social interactions as a whole, over the processing of individual actions [84,85,86]. We also know that perceptual grouping may occur even when part of the stimuli is masked [87]. Likewise, spatiotemporal simulation of a ‘missing part’ might be generated to be compared in real-time with the upcoming input during social interactions [11].

The main limitation of this study is that we attempted to decouple action preparation from action integration and rule out the hypothesis of the selection mechanism: thus, we could not investigate a spontaneous interaction between the agents. Future research directed at investigating the inhibitory mechanisms of joint actions should also include the action integration phase, using paradigms in which response choice is free and without temporal constraints. As a second limitation, one could argue that the inhibition of corticospinal excitability is due to the mere presence of an outstretched hand approaching the participant, without the effect being intrinsically social. We ruled out this alternative explanation in a previous study, in which a moving arrow replaced the social gesture [16]. The results showed that the arrow resulted in much lower MEP activation than the request gesture ([16], see also [72]).

### 4.4. Future Applications

The development of a detailed model of behavioral and neurophysiological correlates of action preparation and execution can help identify the subtle social impairments that characterize motor performance, potentially aiding clinicians in the process of establishing a diagnosis. The analysis of the corticospinal excitability index, as a biomarker of motor function, may provide clinicians with new tools to investigate social deficits in the motor domain. Our preliminary results promote the adoption of the corticospinal index as a promising diagnostic tool. In rehabilitation, the use of ecologically valid actions (i.e., resembling everyday activities) can promote the social and motor recovery of various pathologies characterized by impairments in the motor, inhibitory and social domains (e.g., Parkinson’s disease). Indeed, the social nature of the task would make it more motivating for patients compared to tasks performed in isolation, which is crucial for its implications in rehabilitation.

## 5. Conclusions

We revealed that preparing a motor response after observing an interactive request induces corticospinal inhibition, and this in turn favors a faster and more efficient execution of the action.

Our findings suggest that tightly timed suppression of muscle activity is involved in achieving the elegant choreography characterizing the social interactions of every-day life. As emerges from the present study, the simultaneous recording of several measures holds particular promise for future studies attempting to characterize the neurobehavioral bases of dyadic motor interactions.

## Figures and Tables

**Figure 1 biology-12-00332-f001:**
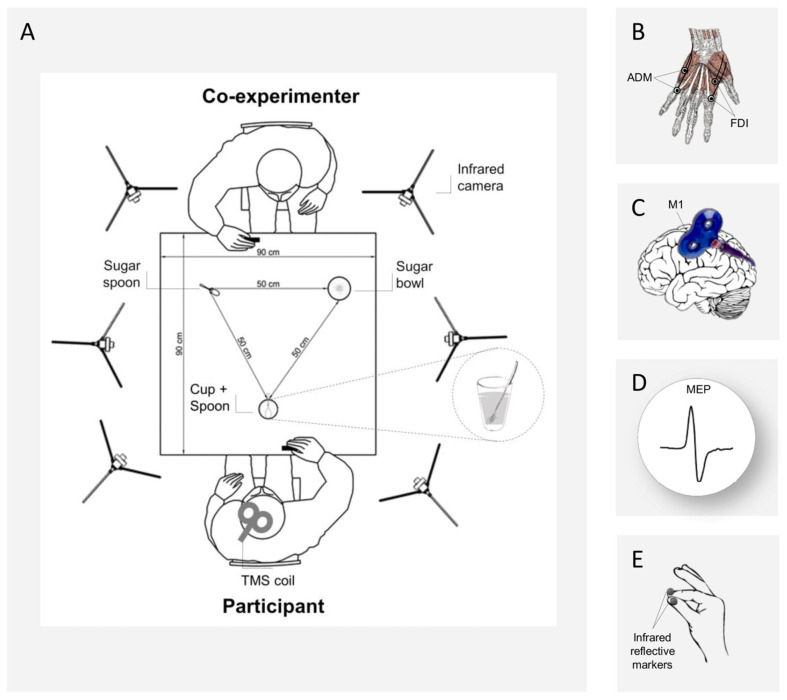
(**A**) Experimental setup. A 3-D Optoelectronic SMART-D system was used to track the participant’s and co-experimenter’s right hands kinematics by means of six infrared cameras. They sat in front of each other on opposite sides of a table. A sugar spoon and a sugar bowl were placed close to the co-experimenter, while a cup full of coffee with a spoon inside was placed close to the participant’s hand. (**B**) The EMG activity of the FDI and ADM hand muscles was recorded through two pairs of surface electrodes. (**C**) A TMS coil placed on the participant’s left M1 was used to investigate corticospinal excitability during action preparation and (**D**) TMS-induced MEPs were recorded. (**E**) Two infrared reflective markers were taped to the participant’s index and thumb fingers and a third was taped on the co-experimenter’s wrist for kinematic recordings.

**Figure 2 biology-12-00332-f002:**
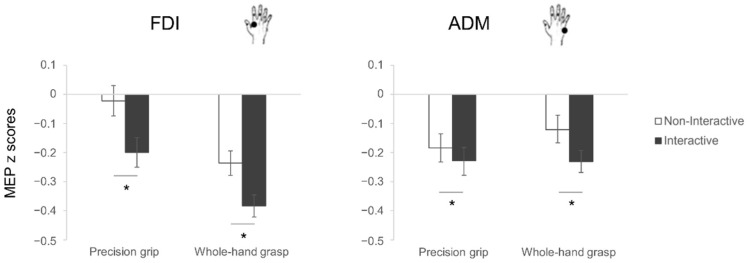
MEP results. Normalized MEP amplitude (z scores) of the FDI (**left**) and ADM (**right**) muscles during action preparation of precision grips (PG, stirring) and whole-hand grasps (WHG, lifting) under Non-Interactive (white) and Interactive (black) conditions. The error bars represent standard error of the mean. Asterisks indicate significant differences (*p* < 0.05) between conditions (planned contrasts).

**Figure 3 biology-12-00332-f003:**
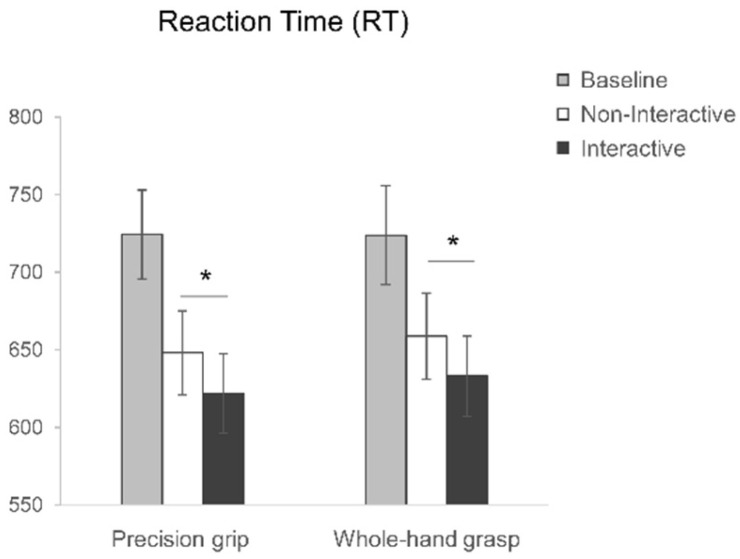
Reaction Time (RT) results for precision grip (PG) and whole-hand grasp (WHG) actions performed in the Baseline (grey), Non-Interactive (white) and Interactive (black) conditions. The error bars represent standard error of the mean. Asterisks indicate significant differences (*p* < 0.05) between conditions (planned contrasts).

**Figure 4 biology-12-00332-f004:**
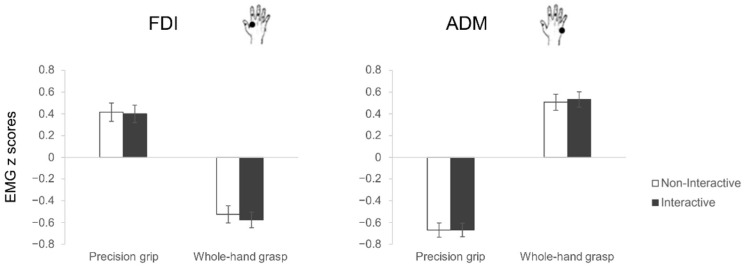
EMG results. The bar charts show EMG activations (z scores) during action execution of precision grips (PG, stirring) and whole-hand grasps (WHG, lifting) under Non-Interactive (white) and Interactive (black) conditions. The error bars represent standard error of the mean.

**Figure 5 biology-12-00332-f005:**
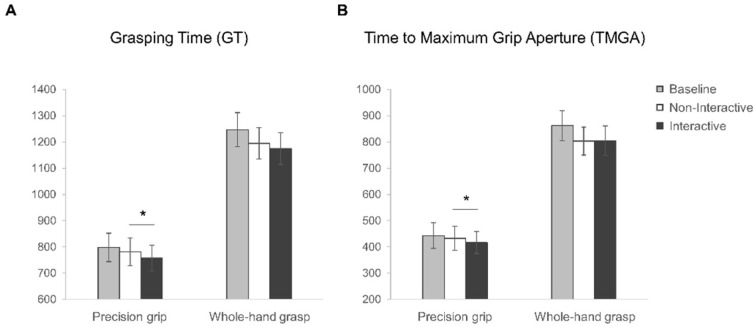
Kinematic results. Graphical representation of (**A**) Grasping Time (GT) and (**B**) Time to Maximum Grip Aperture (TMGA) for precision grip (PG) and whole-hand grasp (WHG) actions performed in the Baseline (grey), Non-Interactive (white) and Interactive (black) conditions. The error bars represent standard error of the mean. Asterisks indicate significant differences (*p* < 0.05) between conditions (planned contrasts).

**Figure 6 biology-12-00332-f006:**
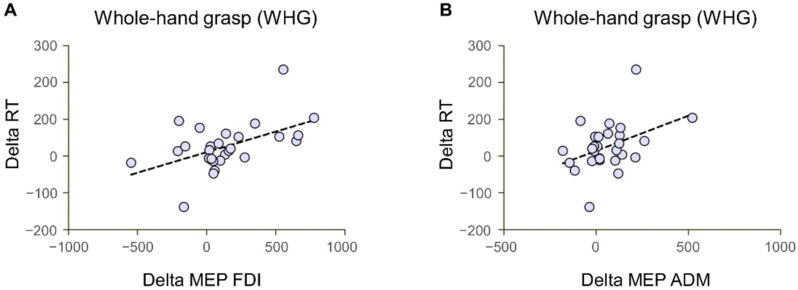
Correlations between MEP and RT delta values (Interactive—Non-Interactive condition). Both the differential index for the FDI (**A**) and ADM (**B**) MEP values positively correlated with the Reaction Times (RT) in the preparation of whole-hand grasps.

## Data Availability

The data presented in this study are available on request from the corresponding author. The data are not publicly available due to data ownership regulations and privacy regulations contained in the informed consent signed by participants involved in the study.

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
