# Peer review of "When Corticospinal Inhibition Favors an Efficient Motor Response"

_biology, 2023, doi:10.3390/biology12020332_

Round 1

Reviewer 1 Report

Dear authors,

Corticospinal inhibition in dyadic motor contexts favors an efficient motor response

Simple Summary

What is the practical use of this corticospinal inhibition?

Abstract

Our results provide new insights on the inhibitory and facilitatory drives guiding social motor response generation – what insights?

Keywords – use mesh terms

Introduction

Is too long

Aim should be better emphasized

Material and methods

Define naïve volunteers

Did you have the Ethical Board approval?

Discussion

The present study confirms and extends previous literature line 501 -which literature?

Numerous researches  - line 528 which?

What is the difference between the previous study with a similar whole-hand grasping task [68] and the current one?

Recent research – which? Line 558

The discussion should be expanded including similar articles like

Bruno V, Fossataro C, Garbarini F. Inhibition or facilitation? Modulation of corticospinal excitability during motor imagery. Neuropsychologia. 2018 Mar;111:360-368. doi: 10.1016/j.neuropsychologia.2018.02.020. Epub 2018 Feb 17. PMID: 29462639.

Siddique U, Frazer AK, Avela J, Walker S, Ahtiainen JP, Howatson G, Tallent J, Kidgell DJ. Determining the cortical, spinal and muscular adaptations to strength-training in older adults: A systematic review and meta-analysis. Ageing Res Rev. 2022 Oct 9;82:101746. doi: 10.1016/j.arr.2022.101746. Epub ahead of print. PMID: 36223874.

Sun Y, Hurd CL, Barnes MM, Yang JF. Neural Plasticity in Spinal and Corticospinal Pathways Induced by Balance Training in Neurologically Intact Adults: A Systematic Review. Front Hum Neurosci. 2022 Aug 17;16:921490. doi: 10.3389/fnhum.2022.921490. PMID: 36061497; PMCID: PMC9428930.

Also, systematic reviews should be considered in the discussion

Limitations of the study should be shown

Conclusion

Should be more specific

First sentence is not a conlsusion

What are the practical tasks?

What are the clinician’s role and future recommendations for corticospinal inhibition in dyadic motor contexts?

Overall recommendations: the manuscript is too long and a little bit confusing

Try to use shorter sentences and reveal originality and practical utility

Reviewer 2 Report

Corticospinal inhibition in dyadic motor contexts favors an efficient motor response
Betti et al. - Biology
In this study, Betti et al. investigate different markers of action observation and motor planning - including TMS-assessed CSE, reaction times (RTs), EMG and movement kinematics - during a real-time social interaction.
1. The report includes many different comparisons and statistical tests (noted; with adequate Bonferroni corrections). Based on my reading however, the experimental contrast of interest is between the interactive and non-interactive conditions, separately per grip action. The reported figures could be restructured to more adequately reflect this (i.e. grouping baseline, non-interactive and interactive data next to each other, structured per grip condition), ideally with an indication of level of significance as reported via the planned comparisons.
2. For some statements in the Discussion, it is ambiguous to which specific outcome variables they refer, and thus unclear how they map onto the obtained statistical results, e.g.:
• "Extending this line of research, here we also observed that in real-time scenarios, reduced MEPs during action preparation were followed by efficient performance during interactive actions execution, in terms of faster and more efficient motor responses." > What is meant by more efficient motor responses? The more since: (i) no differences in EMG activity during actual execution of hand actions between interactive and non-interactive conditions were noted; and (ii) the correlation analysis only demonstrates associations between MEPs and RT's, and no other variables of action execution. This notion is also mentioned in the abstract.
• "We excluded this hypothesis by demonstrating that the co-experimenter always performed the Interactive and Non-Interactive action sequences in the same way, regardless of the subsequent response action." > How was this tested, which part of the data is used to back up this claim?
Note that these two examples are non-exhaustive. Since many different outcome measures are employed, please take care to direct the reader to the corresponding results throughout the Discussion.
3. To me, it is still a bit unclear how the participants' prompted responses (stirring the cup via precision grip or lifting the cup with a whole-hand grip) fit within the interactive request. In my experience (and also in the experience of the participants of the preliminary validation study), neither is a full, socially appropriate response to the request, which would involve lifting the cup towards the co-experimenter (not merely up), who can then pour the sugar in the participant's cup. In this respect, the precision grip and stirring of the cup may be interpreted as an incongruent social response (i.e. ignoring the interactive request), whereas the whole-hand cup lifting may be considered a (partially) congruent response (i.e. lifting the cup to facilitate the requested sugar pouring) (although again, it can be argued that both may be considered incongruent in a daily social setting). I think this social congruence/incongruence of the prompted action is an important conceptual factor here, which is not reflected in the analysis nor interpretation of the obtained results.
4. The comment above could maybe also explain why correlations between preparatory MEPs and action-related outcome measures (RTs) were only observed for the whole-hand grip, not the precision grip (i.e., only for the socially congruent response). Perhaps the authors have another explanation for this finding, which I’d be happy to read in the Discussion.
5. Eye contact is known to be an important factor for conveying interactive requests, and is also known to modulate CSE, as shown in earlier work from the same group of authors. Was this factor taken into account and/or standardized across conditions? Or could the differences encountered in the interactive vs. non-interactive condition be due to differences in eye contact by the co-experimenter?

6. Minor comments:
─ Page 8 final paragraph: when reporting statistics, sometimes the 'p =' is missing.
─ To increase readability, preference to write precision grip and whole-hand grip always in full, instead of the PG and WHG abbreviations, also in figures.
─ Figures could be grouped more thematically, i.e. following the in-text descriptions of results distinguishing between outcome measures related to motor preparation vs execution.
─ Figure legend 3: "RT (A), GT (B) and TMGA (C) for PG (white) and WHG (black) actions..." is a bit unreadable with all abbreviations.
─ Similar comment about the figure axes in figure 4.
─ Preference to include the exemplary pictures of the experimental setup, now in the supplementary materials, in the main manuscript.

Round 2

Reviewer 1 Report

The manuscript has been improved 

Author Response

  1. The manuscript has been improved.

R1. We thank the Reviewer for the help in improving the overall quality of the manuscript. Based on the comments received from all reviewers, in this second revision we tried to further clarify the theoretical framework of the study as well as its methodological details and the description of the results. We hope that the current version of the manuscript can match the Reviewer’s expectations.

Reviewer 3 Report

It seems that the authors underestimated the criticisms I made in the first review of the article. It is not just a question of removing a few words from their manuscript but of totally modifying the setting of the manuscript. In fact, what they can argue is that greater inhibition of the CSE system facilitates the execution of a predetermined response, when this must be performed in response to an irrelevant stimulus that evokes a different response. They cannot say anything about joint goals, social engagement or social context. They may refer to these concepts only in discussion if they wish.

Here are some further comments that may, I hope, clarify the criticism better:

Summary

I still disagree with the summary: “With this neural finding, we shed light on the critical intermediate phase of motor processing between the observation of the action and the action integration phase aimed at achieving of a joint goal”. The authors assume that their findings can be generalized to a situation where joint action is actually executed. Their experimental protocol does not allow for this. To state this it is necessary to have a condition in which the experimenter and the participant interact.

Abstract

Again: “Participants were requested to perform an action (i.e., stirring the coffee or lifting a coffee cup) following a co-experimenter’s request gesture. A non-interactive condition was also included.” None of the experimenter's actions are interactive or non-interactive. The description is still ambiguous for the reader. It is possible to describe situations as: irrelevant stimuli that may evoke an automatic response vs neutral irrelevant stimuli. The fact that they are actions is incidental. As written below, even non-biological stimuli can evoke automatic responses. If the authors want to persist with their interpretation of the experimental situation they must add control conditions in which a non-biological stimulus evokes an automatic response and they must find differences between the biological situation (action) and the non-biological one (e.g. falling object).

Introduction

Line 113: “a byproduct of increased preactivation due to social engagement”: the preactivation could have occurred even if an object had fallen from above or thrown towards the subject who had the instruction not to move. It is not possible to generalize the situation to something exclusively "social".

Line 114: “if the observation of an interactive request strongly triggers the preparation of an appropriate response action, which, however, must be postponed, then more inhibition is required”: more inhibition with respect to what? Why? Please, add some bibliography.

Line 117: “i.e. to perform the action appropriate to the social context”: this sentence must be something similar to “i.e. to perform the action automatically evoked by the stimulus”

Experimental paradigm

Line 146: I don't understand the choice of using the terms request or action in the different experimental situations. What is resolved?

Line 180: what is meant by social appropriateness?

Round 3

Reviewer 3 Report

I remain convinced that the experimental protocol used, which requires participants to perform a predefined response depending on the experimental block, is not adequate for the experimental question. Consequently, the discussion does not adequately consider the results obtained.

The term "irrelevant" attributed to the observed action is a technical term in experimental psychology attributed to any stimulus that does not require processing to perform the task. Irrelevant stimuli may interfere with the task or modulate the response. However, they remain irrelevant to the task.

Considering the boundless literature that has used this type of experimental protocol, I repeat, the authors can only argue that greater inhibition of the CSE system facilitates the execution of a predetermined response, when this must be performed in response to an irrelevant stimulus that evokes a different response. A result that I find interesting.

Any interpretation attributing a specific effect to the "social content" of the irrelevant stimulus must be supported by an adequate control experiment.